# Controversies and Clinical Applications of Non-Invasive Transspinal Magnetic Stimulation: A Critical Review and Exploratory Trial in Hereditary Spastic Paraplegia

**DOI:** 10.3390/jcm11164748

**Published:** 2022-08-14

**Authors:** Rafael Bernhart Carra, Guilherme Diogo Silva, Isabela Bruzzi Bezerra Paraguay, Fabricio Diniz de Lima, Janaina Reis Menezes, Aruane Mello Pineda, Glaucia Aline Nunes, Juliana da Silva Simões, Marcondes Cavalcante França, Rubens Gisbert Cury

**Affiliations:** 1Department of Neurology, School of Medicine, University of São Paulo, São Paulo 05403-000, Brazil; 2Department of Neurology, University of Campinas, Campinas 13083-888, Brazil; 3Instituto de Ciências Matemáticas e de Computação, University of São Paulo, São Carlos 13566-590, Brazil

**Keywords:** magnetic stimulation, spinal cord, neuromodulation

## Abstract

Magnetic stimulation is a safe, non-invasive diagnostic tool and promising treatment strategy for neurological and psychiatric disorders. Although most studies address transcranial magnetic stimulation, transspinal magnetic stimulation (TsMS) has received recent attention since trials involving invasive spinal cord stimulation showed encouraging results for pain, spasticity, and Parkinson’s disease. While the effects of TsMS on spinal roots is well understood, its mechanism of action on the spinal cord is still controversial. Despite unclear mechanisms of action, clinical benefits of TsMS have been reported, including improvements in scales of spasticity, hyperreflexia, and bladder and bowel symptoms, and even supraspinal gait disorders such as freezing and camptocormia. In the present study, a critical review on the application of TsMS in neurology was conducted, along with an exploratory trial involving TsMS in three patients with hereditary spastic paraplegia. The goal was to understand the mechanism of action of TsMS through H-reflex measurement at the unstimulated lumbosacral level. Although limited by studies with a small sample size and a low to moderate effect size, TsMS is safe and tolerable and presents consistent clinical and neurophysiological benefits that support its use in clinical practice.

## 1. Introduction

Magnetic stimulation uses strong currents through coils to generate large and rapidly changing magnetic fields that induce electrical currents that stimulate deep tissue, depolarize membranes, and alter neuronal excitability. Since its initial development in 1985, magnetic stimulation has been demonstrated to be well tolerated and has shown great promise for the treatment of neurological and psychiatric disorders via stimulation or inhibition of brain networks [1]. Transcranial magnetic stimulation (TMS) has dominated the literature, but stimulation of other tissues was also explored considerably in the early years of the technique, including stimulation of peripheral nerves, peripheral magnetic stimulation [2], and stimulation over the spinal column, transspinal magnetic stimulation (TsMS) [3].

Several terms are used for magnetic stimulation at the spine and are often interchangeable throughout the literature. Cervical or lumbar root magnetic stimulation centers on peripheral spinal root stimulation and is preferably employed by neurophysiologists, sometimes receiving the term peripheral magnetic stimulation (PMS). Another term is “transcutaneous magnetic spinal cord stimulation”, which presumes spinal cord stimulation. In this review, “TsMS” is used, as it better describes the method uniquely in its direct action and does not automatically include or exclude putative central nervous tissue action. 

Interest in spinal cord stimulation has gained traction since promising results were obtained in clinical experiments with invasive stimulation in Parkinson’s disease (PD) [4], motor recovery after spinal cord injury [5], and innovations in pain control [6]. Renewed interest in clinical applications of TsMS has arisen from the exciting possibility of offering similar neuromodulation with non-invasive alternatives combined with favorable safety and cost profiles. There is, however, controversy about whether the mechanism of action relies on direct spinal tissue modulation or indirect peripheral pathways, as unlike TMS, which can reliably produce compound muscle action potential (CMAP) in anatomically respective muscles [1], TsMS appears to only produce CMAPs in muscles whose roots are close to the coil with little evidence of intraspinal tract activation [7].

Previous authors explored spinal excitability and documented possible modulatory properties of TsMS through the H reflex, an electrophysiological technique consisting of a reflex reaction of a muscle after stimulation of Ia afferents [8]. The H reflex recovery curve, obtained by applying sequential pairs of conditioning stimuli at increasing intervals, can also assess spinal circuitry with early facilitation documented in hyperexcitability [9]. Spasticity research gives us further insights into the mechanisms underlaying spinal hyperexcitability. Abnormalities in propriospinal circuits that control reflex pathways may lead to an increase in reflex excitability in spastic patients with spinal impairment [9]. In this scenario, patients with pure spastic paraplegia could provide an ideal opportunity for the evaluation of the modulatory effect of TsMS. 

In this review, we explore the rationale behind the technique and past exploratory studies and discuss current therapeutic strategies. Additionally, we explore neurophysiological changes induced by TsMS in spastic paraplegia patients to expand the understanding of its mechanism by addressing whether the technique can modulate a spinal segment distal to the stimulation site. 

## 2. Materials and Methods

We performed a comprehensive critical review of the literature. Two authors, neurologists with experience in neuromodulation, independently searched Medline, Embase, and Google Scholar for the terms “Magnetic stimulation” in combination with either terms “transspinal”, “Spinal cord”, “cervical”, “Thoracic”, or “Lumbar”. The last search was run on 18 July 2022. We did not restrict the language or date of publication. Two authors independently included all exploratory or clinical studies involving magnetic stimulation performed over the vertebral column in humans with no regard for outcome. Additionally, the reference and citation lists of relevant studies were manually screened for more eligible articles. Exploratory studies in animal or computational models were selected to discuss the mechanism of action of TsMS.

For didactic reasons, we organized our manuscript to answer two questions:

How does TsMS work?

What are the clinical implications of TsMS? 

Next, we present an exploratory pilot study including three patients with hereditary spastic paraplegia (HSP) who were subjected to TsMS in order to assess the neurophysiological response in spinal cord regions not directly affected by stimulation. The inclusion criteria were age >18 years and diagnosis of pure HSP based on a genetic testing. Subjects with comorbid neurological or orthopedic disorders were excluded. The H-reflex recovery curve [9] was applied at baseline and immediately after the completion of the stimulation protocol consisting of two continuous theta burst TsMS sessions. 

The TsMS was performed with participants seated comfortably in a quiet room at an ambient temperature of protocol at 25 °C. Stimulation was applied using a MagVenture^®^ MagPro R20 equipment (Alpharetta, GA, USA) with a figure-eight coil at the level of the second thoracic vertebra, avoiding lumbar and sacral roots activation. The coil was placed perpendicular to the spinal cord. The stimulation was set at 100% of the motor threshold for diaphragm contraction, by gradually increasing stimulation intensity, while a second examiner monitored for muscle contraction. Each continuous theta burst session consisted of a hundred three-pulse bursts at 50 hz for a total of 20 s, and the session was repeated with an interval of 90 min. 

Neurophysiological evaluations were performed with the participants laying on a stretcher in a room with the same conditions. Considering that the participants had symmetric motor impairment in both legs, the recordings and data analysis were carried out only in the left side for minimizing discomfort. For electrophysiological recordings, we used disposable electrodes (Ambu repositionable solid gel electrodes) placed over the soleus muscle. 

All exams were performed with a Neurosoft^®^ Neuro-MEP-Micro electromyograph (Ivanovo, Russia). The gain was set at 2 mV/division, and the band-pass filter frequencies were 2 Hz to 10 kHz. H-reflex recovery curves were obtained by paired stimuli at interstimulus intervals of 50, 100, 150, 200, 250, 300, 350, 400, and 500 ms. Five traces were recorded for each paired stimulus with a pause of at least 15 s between 2 consecutive stimuli. The evaluation complied with the Good Clinical Practice guidelines and the International Organization for Standardization. It was approved by the FMUSP institutional review board, and all participants signed an informed consent form before any study-related procedure was performed.

## 3. Results

### 3.1. How Does TsMS Work?

Rodent models have been used extensively for evaluating the effects of TsMS on spinal cord tissue. TsMS causes axonal depolarization, associated glutamate release, and increased excitability [10]. Furthermore, in rodents, TsMS activates the corticospinal tract, leading to lower limb CMAP [11]. TsMS may also have a possible effect on spinal injury by promoting neuronal and functional recovery through axonal regrowth and increased neuronal survival on mice models [12]. Animal models have also been used for addressing safety early concerns about the induction of magnetic fields in the proximity of the heart, phrenic nerves, and sympathetic ganglia. Magnetic fields of up to 2.4 Tesla do not induce cardiac arrhythmias, even after sensitization with digitalis [3]. 

Most exploratory studies in humans were conducted in the early years of magnetic stimulation, summarized in Appendix A. Several authors identified that produced CMAP has a fixed latency regardless of the coil position or intensity, even when moving the coil away from the stimulation site in any direction laterally or rostro caudally up to the point where it is no longer elicitable [13,14,15], and conclude that stimulation would occur at a fixed point for each studied muscle, and therefore, unlikely to be from direct corticospinal tract activation and possibly at an area of high current concentration such as the neuroforamina. Additionally, CMAP latencies were found to be always shorter than F-wave latencies used to measure latency in the anterior horn neuron [16,17] and are comparable to transcutaneous electrical simulation focused at the root exit from the spinal column [13]. A single study by Cros and colleagues in 1990 indicated conflicting results in latency studies suggesting stimulation sites at the anterior horn or at the motor roots immediately exiting the spinal cord [18]. However, such results have not been replicated. 

Claude Tomberg performed the only TsMS study supporting descending tract activation in 1995, describing that stimulation at C7 led to the production of CMAPs compatible with spinal root activation, whereas when the coil was placed 6 to 8 cm higher, CMAPs with up to 2.5 ms longer latencies were obtained. This led to the conclusion that multisynaptic motor activation may have been achieved by either corticospinal or propriospinal systems [19]. The high cervical position of the circular coil was explored by another group of authors, aiming, however, for foramen magnum level stimulation. Using a double-cone stimulator centered at the inion or high cervical levels directly below, Ugawa demonstrated the possibility of corticospinal stimulation in seven healthy volunteers in 1994 [20]. CMAPs had constant latencies indicating a fixed stimulation site, which was estimated to be at the foramen magnum considering the concentration of the induced field at bony canals. A comparison of electrical brainstem stimulation with this technique was performed along with collision studies, documenting the main component in elicited stimulation to be the large-diameter component of the corticospinal tracts, further localizing the stimulation site to just below the pyramidal decussation. While this technique is useful for diagnostic purposes in accessing central conduction times in conjunction with TMS [21] and TsMS [22], stimulation would not occur at cervical spinal cord, and thus, would not clearly characterize evidence of TsMS action on descending tracts.

Structures other than motor roots were demonstrated to be stimulated. Although no paresthesia is produced, somatosensory evoked potential has been used to confirm that sensory fibers could also be stimulated with TsMS, probably between the dorsal ganglia or spinal foramen [23]. The phrenic nerve trunk has been reliably stimulated and led to no adverse effects [24], but sympathetic cervical ganglia could not be stimulated; thus, adversely demonstrating that cervical stimulation with 3.5 Tesla leads to no significant cardiovascular changes or side effects [25]. 

The most recent study to explore TsMS mechanisms in humans was published in 2021 and had similar findings to previous literature. The components of TsMS-elicited CMAP were described, and the excitatory and inhibitory responses were evaluated. The study concluded that all findings were compatible with mixed nerve stimulation at very proximal sites [26]. There was no descending tract activation when using up to 200% of the resting motor threshold. 

The obstacles hampering adequate spinal cord tissue activation by TsMS have been subjected to scrutiny by exploratory studies. Maccabee and colleagues explored the effect of bone geometry on magnetic stimulation using a human vertebral column in saline, demonstrating that the maximum generated current occurred at the neuroforamina, corroborating the fixed point of stimulation found in electrophysiological studies, while generated current was minimal at the center of the spinal canal. Coil type and positioning determined current intensity at both sites. The highest generated longitudinal current was found to be induced from paravertebral centering of a circular coil [27]. Additional work with computational modeling explored the current distribution with a double 70-mm coil, suggesting a heterogeneous current distribution with increased concentration in dorsal columns and dorsal horns, while sparing the anterior horns and anterolateral columns. While the calculated current would be approximately a tenth of previously measured at the neuroforamina under TsMS, it is still predicted to be 30 to 60 times higher than that in conventional direct current stimulation. When generating longitudinally oriented currents, neurons oriented in left-right directions are favorably affected, favoring dorsal root and interneurons, while transversely oriented current preferably affect longitudinal neurons, favoring dorsal column neurons [28]. Another simplified computational model was used to estimate the energy required for generating action potentials in spinal cord tracts, resulting in more than 35 times the maximum voltage for a conventional 70-mm figure-of-eight coil [29], suggesting that a conventional TMS device would not be ideal for this purpose.

In vivo animal studies have allowed for further exploration. In an experiment with 10 dogs, spinal cord descending tracts could only be stimulated by concomitant laminectomy and pediculectomy [30], which is a practical demonstration of how bone hampers magnetic induction. In 14 pigs, the technique failed to stimulate descending tracts even after the same extensive bone resection, possibly due to the depth of the spinal cord in the particular animal [31]. Both studies documented reliable and robust CMAPs produced at muscles innervated by roots adjacent to the TsMS coil. They also demonstrated the stimulated structure to be the motor root by performing spinal cord sectioning and proximal root rhizotomy, with no change whatsoever in observed CMAPs. 

Some studies have delved into electrophysiological changes provoked by TsMS and demonstrated spinal cord circuitry changes. Nielsen and Sinkjaer further explored the mechanism of improvement of spasticity following TsMS by investigating 20 volunteers (including 11 patients with multiple sclerosis and 9 healthy controls). These authors concluded that mid-thoracic TsMS inhibited soleus motor neuron excitability. The H-reflex depression lasted longer in healthy controls and with 25 Hz than with 1 Hz, but it occurred in both cases despite the usual opposite cortical neuromodulatory effects. The H-reflex depression was heterogeneous from cervical stimulation and more pronounced when applied directly over soleus roots [32]. Knikou et al. described a reduction of the soleus H-Reflex after sequentially evoking CMAPs in lower-limb muscles (including the soleus) at conditioning intervals reaching 0.2 Hz with a figure-eight coil centered at T9–10. The exact mechanism of H-reflex depression was unclear but was attributed to possible Ia afferent hyperpolarization through antidromic excitation of group Ia afferents at the same level of stimulation [33].

### 3.2. What Are the Clinical Implications of TsMS?

Despite the mentioned controversial mechanisms, TsMS has been used in several clinical studies with the hope of producing beneficial effects in motor and autonomic functions, namely spasticity, autonomic functions, and Parkinsonian symptoms. These are summarized in Table 1.

Spasticity has been the initial focus of TsMS. Three trials were done by the same authors using 12-Hz and 25-Hz rTsMS at eight thoracic vertebrae in multiple sclerosis patients with spsasticity, including a 38-patient double blinded placebo-controlled trial. This led to reduced clinical measures of spasticity on the Ashworth scale and reflex grading scale. Benefits in activities of daily living were seen in the pilot study, but were superior in the sham group in the randomized controlled trial [32,34,35]. Another author used a coil centered 2 cm laterally for lumbar stimulation in three trials, totaling 22 patients with spinal cord injury, multiple sclerosis, familial spastic spinal paralysis, transverse myelitis, and spinal vasculitis, as well as 16 healthy controls. The results were similar with clinical reduction of the Ashworth and reflex grading scales [36,37,38]. Interestingly, the results were more pronounced contralaterally to the stimulated side and the greatest clinical benefit occurred in 4–24 h (not immediately after), decaying after 48 h after one stimulation [37]. This study aimed at spinal root activation, but as previously discussed, a paravertebral approach would still lead to current induction inside the spinal canal, although the effect on spinal roots would still be greater. The fact that the contralateral side showed greater spasticity reduction suggests spinal cord modulation. A subsequent analysis of optimal frequency of repeated stimulation between 10 Hz, 15 Hz, and 20 Hz failed to determine the best setting. All parameters were inhibitory, and the study had only one subject, but all results were significantly superior to those of a sham stimulation [38].

Spasticity is associated with increased stretch reflexes and increased amplitude of the H-reflex, which can be used as biomarkers to assess spasticity [45]. Nielsen et al. and Knikou et al. documented decreased amplitudes of the stretch reflex, an increase in the threshold of the stretch reflex, and a decrease of the H-reflex, supporting the clinical improvement described [32,33]. 

For autonomic dysfunction, transcutaneous magnetic stimulation led to neuromodulation of spinal-micturition circuitry and improved the bladder function in neurogenic bladders associated with spinal cord lesion in a proof-of-concept study with five patients [39]. A weekly dose of TsMS at 1 Hz led to improvement in the voluntary urination volume and the frequency of catheterizations after 16 weeks. Additionally, a benefit was documented in urodynamic testing, which revealed an increase in bladder capacity and urinary velocity. One man also reported a benefit in sexual disfunction. When patients started the sham phase, the benefit decayed. Aiming to define optimal settings, the study also compared 1 Hz and 30 Hz protocols, leading to an improvement in urodynamic parameters only in the former.

Anorectal function may also be improved by TsMS, shown to benefit the bowel program for patients with neurogenic bowel. Two unblinded studies by the same group applied TsMS to T9 and L3 guided by 80% of the threshold to induce lower-limb muscle contraction in twice daily sessions for 21 days in 38 patients with complete and incomplete spinal cord injury and Parkinson’s disease with refractory constipation. The clinical benefit documented in both trials was a reduction in colonic transit time and improvement in bowel symptoms according to questionnaires [40,41]. Response was similar in patients with complete and incomplete spinal injury. 

TsMS has been used for Parkinson’s disease. In a small open-label study with five patients with freezing of gait, intermittent theta-burst TsMS at T5 improved the freezing of gait questionnaire scores and UPDRS score after one week of stimulation, but not immediately after TSMS or after 28 days [43]. Another trial used repetitive TsMS at 5 Hz provided immediate camptocormia improvement in a single-blinded crossover trial with 37 participants, with the effect lasting for reportedly around three days, and no difference from baseline observed after one week. The mechanism proposed in both studies is that the spinal cord is stimulated in a manner similar to epidural spinal cord stimulation. Epidural spinal cord stimulation has been proposed as a possible therapeutic approach through dorsal ascending pathways and upper modulation of the brainstem nuclei, thalamus, and cerebral cortex. The aim in this case is to disrupt aberrant oscillatory activity in the beta range [46]. 

The largest study using TsMS to date investigated benefit on symptoms and signs of parkinsonism in patients with moderate Parkinson’s disease [44]. In a single blinded randomized clinical trial in 100 patients with moderate severity (Hoehn and Yahr 3 and 4), eight sessions of 5 Hz repetitive TsMS at T12/L1 level and multidisciplinary rehabilitation was significantly superior to rehabilitation alone in improving the Unified Parkinson’s Disease Rating Scale and Time Get and Go Test scales for parkinsonism and gait function. The benefit was greater after TsMS but was still significantly present after 6 months. The study was hampered by a high dropout rate but similar in both groups (34–36%), and while sham was used, the authors considered the patients unblinded due to intense paraspinal muscle contraction only present during active TsMS. 

TsMS is also useful as a diagnostic tool when combined with surface electromyographic recording, cortical stimulation, or foramen magnum stimulation [25], thus integrating the analysis of motor root stimulation at spinal levels. The analysis of latencies at all levels enables the evaluation of central motor conduction above and below the foramen magnum and complements peripheral conduction studies. While transcutaneous electrical stimulation can be used for the same purpose, it also stimulates small nerve fibers in subcutaneous tissue and the skin, causing significant pain [26].

### 3.3. Open-Label Pilot Trial

Three patients with hereditary spastic paraplegia who presented pure phenotypes were selected. Their ages were between 37 and 63 years, and two were female; they had no fixed muscle contractures and had spastic paraplegia rating scale baseline scores between 21 and 34, with no use of medication. All three patients completed all evaluations and reported no adverse effects or discomfort from TsMS. No lower limb muscle contraction or sensory disturbance were observed with any intensity. Patient characteristics and the results of the H reflex recovery curve are summarized in Table 2 and Figure 1. 

After 100 ms, the conditioned H-reflex returned to the baseline in all patients. Patient 3 could not complete the test after 150 ms. At 150 ms pre and post stimulation, patient one had a H_2_/H_1_ ratio of 1.13 and 0.96, patient two had a ratio of 1.01 and 0.98, and patient three had a ratio of 1.06, pre stimulation only. All H_2_/H_1_ ratios after 200 ms in both pre and post TsMS were 1.

## 4. Discussion

Clinical evidence for TsMS is limited but promising. We found a consistent clinical benefit in all explored pathologies involving different functions of the spinal cord, with associated biological plausibility due to response in neurophysiological biomarkers, as summarized in Table 3. Its use in Parkinson’s disease was not supported by biomarkers, but those studies benefit from larger number of participants and controlled designs. 

No conclusion can be obtained regarding optimal parameters, as stimulation protocols are varied, heterogeneous in either inhibitory or excitatory paradigms, number of pulses and sessions, coil design, and intensity, and decided on the basis of individual researchers’ reasoning. While TMS sessions produce reliable cortical effects that outlast the initial stimulation period, either inhibitory, with low-frequency stimulation (1 Hz or less) or possibly continuous theta burst protocols, or excitatory, with high-frequency stimulation (5 Hz or higher) and intermittent theta burst protocols [47], there is not yet enough data supporting the validity of the same protocols for the spinal cord. Direct comparisons are limited, but H reflex was shown to be reduced in both low and high frequencies [33].

The fact that only positive studies were found in this comprehensive review instills some optimism, but we cannot rule out a publishing bias at this moment. Moreover, current studies are limited by insufficient correction for confounding factors, heterogenous participants, lack of blinding, and overall small sample sizes. More data are required before the technique can be implemented in clinical practice.

Taken together, the clinical evidence gathered is highly suggestive of spinal cord modulation, but the mechanism is unclear. Unlike TMS, TsMS is incapable of descending tract stimulation with conventional devices; therefore, it is reasonable to question how spinal cord tissue can be modulated by stimulation.

As detailed by exploratory studies, the main obstacle for spinal tissue activation is the bone structure, as high impedance structures are capable of altering the intensity and distribution of the induced magnetic field [48]. TMS devices and coils have been designed to accommodate the requirements for surpassing scalp, bone, and meninges, and stimulate cortical structures. Therefore, the use of the same devices for TsMS with the goal of surpassing vertebrae and stimulating the spinal cord directly might simply not be possible. Current distribution was thoroughly explored by previous work, detailed in Figure 2, and documented clear current concentration in the foramina, leading to easily and reliably spinal roots stimulation, as also documented by CMAPs obtained from TsMS. 

The first and most viable possible mechanism is the indirect modulation of the spinal cord through sensory roots stimulation involving Ia afferents, as suggested by Knikou [33], or antidromically through motor root stimulation. In the latter case, it would indirectly inhibit hyperexcitable motoneurons, which would logically occur at the same spinal level as the stimulated roots. Comparatively, transcutaneous electrical stimulation over the spinal column, a technique that also preferentially stimulates spinal roots, has also been shown to reduce the amplitude of the H-reflex after double pulses and conditioning-test, resembling what happens after TsMS [49], and further suggesting that the main mechanism could depend on root stimulation. It has to yet be determined what effect a massed and rhythmic afferent and efferent root activation could produce apart from isolated root activation.

Another possibility would be that the intense direct muscle recruitment and peripheral nerve stimulation at the stimulation sites, undoubtedly a limited form of peripheral magnetic stimulation, would indirectly modulate spinal tissue. When performed on limbs, it has been documented as being able to reduce spasticity scores and alter tendon reflexes with several inhibitory protocols, including 5 Hz, 20 Hz, 50 Hz, and continuous theta burst [50]. While this could be initially attributed to a proprioceptive modulation of upper spinal circuitry, H-reflex and F-wave were left unchanged in two studies that included neurophysiological testing [51,52]. This favors a more direct effect of peripheral magnetic stimulation on neuromuscular propagation, altering fast-twitch unit fibers and no direct change in spinal excitability, and therefore, this would not entirely explain the TsMS mechanism.

Finally, some current is still generated and apparently concentrated at the dorsal column and dorsal horn, away from descending corticospinal tract or more noticeable sensory spinothalamic pathways [28]. There is a possibility that this slight current, especially if employed in repetitive stimulation protocols, would be able to alter spinal interneurons or alpha motor neurons excitability, but this is not clear at this moment. Specialized hardware could allow unequivocal spinal cord modulation via magnetic stimulation with higher energy and specialized coils, possibly stimulating descending tracts and producing definite proof of overcoming the vertebral anatomy. A 200 mm circular coil developed for cauda equina stimulation could fit this role [53], but the unproven safety of such devices over thoracic or cervical regions must first be addressed.

For Parkinson’s disease, an additional possible mechanism relies upon the described concentrated current on dorsal columns, where TsMS stimulation could provoke an ascending disruptive stimulus in proprioceptive pathways through firing rate entrainment. This is hypothesized to be able to modulate the brainstem and basal ganglia networks and reduce beta hypersynchronization, which is akin to results in animal studies [54]. 

Not only the mechanism of TsMS spinal cord modulation is not clear, but our review highlighted a significant question, as spinal cord assessment via H reflex was mostly analyzed at the level of stimulation. Not only could peripheral mechanisms be more involved than spinal circuitry itself, but no information was obtained from different spinal levels. In a recent study with an animal model of spinal injury, 10 Hz TMS was able to not only to improve motor function, but promote upregulation of pro-inhibitory membrane proteins and reduce the H reflex compatible with reduced motoneuron hyperexcitability [55], suggesting that spinal cord modulation is possible by descending modulation, at least from the cortex. We conducted an open label trial to address if this could also be possible from higher spinal levels using TsMS.

Known patterns of facilitation of H-reflex recovery have been described in individuals with spasticity secondary to stroke and spinal cord injury [56,57]. They present a facilitation of H-reflex recovery when compared to healthy subjects, a phenomenon attributed to corticospinal tract hyperexcitability [8,58,59]. Our findings in the pilot trial showed that the patients with spastic paraplegia also present such hyperexcitability, confirmed by facilitation of the H-reflex to paired stimuli. After the inhibitory TsMS protocol is applied at the level of the second thoracic vertebra, conditioned soleus H-reflex responses were almost completely inhibited at interstimulus intervals of 50 ms, but then returned to a state of excitability at 100 ms in all patients, suggestive of reduced excitability of spinal circuitry. As stimulation at the upper thoracic level would not be able to reach lumbosacral roots [26], our trial documented effects not limited to spinal levels directly under the coil suggests that TsMS is able to modulate the spinal cord at distant levels. Propagation could be through descending corticospinal modulation, either by direct or indirect means at the site of stimulation. 

These results suggest that our continuous theta burst TsMS protocol may have antispastic effects, but the clinical implementation of these hypotheses need validation in further studies. Additionally, both TsMS and H-reflex protocols were safe and tolerable, and no adverse effects were reported.

Considering safety, all experiments published so far using conventional coils documented the lack of serious adverse effects from therapy. A total of 222 neurological patients and 199 healthy controls were stimulated in the reviewed exploratory and clinical literature. Some patients reported mild lumbar discomfort or fatigue [42,43,44], which disappeared by the next day. Other patients reported a feeling of wearing a narrow ring around the mid-thoracic level [34]. Finally, five patient reported brief dizziness, and a single patient reported palpitation after two hours of stimulation, with normal electrocardiogram [34,35,44]. Overall, we found no evidence suggesting any serious risk from TsMS, but no data are available on patients with pacemakers, epilepsy, intrathecal infusion devices, severe vertebral disease, spinal compression or stenosis, previous orthopedic surgery, or any other implanted devices at or in close proximity to stimulated regions. At this point, one should be careful about using TsMS for patients with these comorbidities. Moreover, safety over 1.5 Tesla is less clear, as very few studies employed intensities over this value.

While TsMS is relatively inexpensive, easy to perform, reliable, and safe, its use in future trials must account for two very significant limitations. Currently, there are no clear stimulation protocols, as most are adapted from TMS. There is not yet enough evidence to guide the number and length of sessions or to indicate whether classical cortical inhibitory or excitatory frequencies can be applied to spinal cord modulation. Additionally, there is always intense muscular recruitment in paraspinal muscle, which is extremely noticeable for the patient and makes blinding more difficult. Double-arm protocols can use other forms of distraction involving superficial muscle recruitment, such as transcutaneous electrical nerve stimulation, but crossover protocols must be carefully planned or even reconsidered.

In conclusion, TsMS is a safe technique with current routine use in spinal root stimulation for central conduction time assessment complementing electrophysiological studies. Unlike TES, which can be used for the same purpose, it has a clear advantage of being painless, although with much less precision in stimulation. TsMS cannot stimulate descending tracts in the spinal cord with current devices, and CMAPs obtained from stimulation are mainly due to motor root activation. Besides, TsMS appears to be able to modulate some spinal circuits not only at the level of stimulation. The mechanisms underlying these effects are poorly understood at this point. Despite that, TsMS has been explored as a potential therapy not only for primary spinal cord-related manifestations, such as spasticity, neurogenic bladder, and bowel disorders, but also Parkinson’s disease in alleviating parkinsonians signs and symptoms as well as specific features as freezing of gait and camptocormia. While initial results are yet to be confirmed in larger and adequately blinded trials, and stimulation protocols are yet to be standardized, TsMS is a very promising technique for neuromodulation.

## Figures and Tables

**Figure 1 jcm-11-04748-f001:**
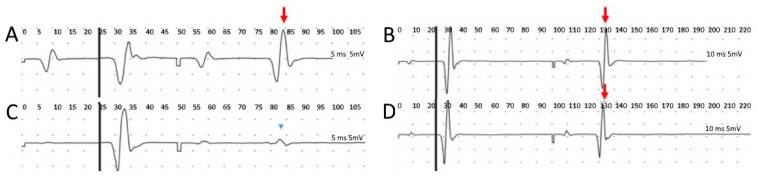
The patient 1 recovery curves of H-reflex obtained by delivering paired stimulus at interstimulus intervals (ISI) of 50 ms and 100 ms, respectively, pre (**A**,**B**) and post (**C**,**D**) TsMS inhibitory protocol. Note that in figure (**A**), before inhibitory modulation with TsMS, the conditioned H-reflex was present and with high amplitude at ISI of 50 ms (arrow). This provides neurophysiological evidence of neuronal hyperexcitability in this patient. After the TsMS inhibitory protocol, the conditioned H-reflex at 50 ms was almost completely inhibited (arrow head) (**C**). There was no evident difference in conditioned H-reflex with ISI of 100 ms before (**B**) and after (**D**) neuromodulation (arrow).

**Figure 2 jcm-11-04748-f002:**
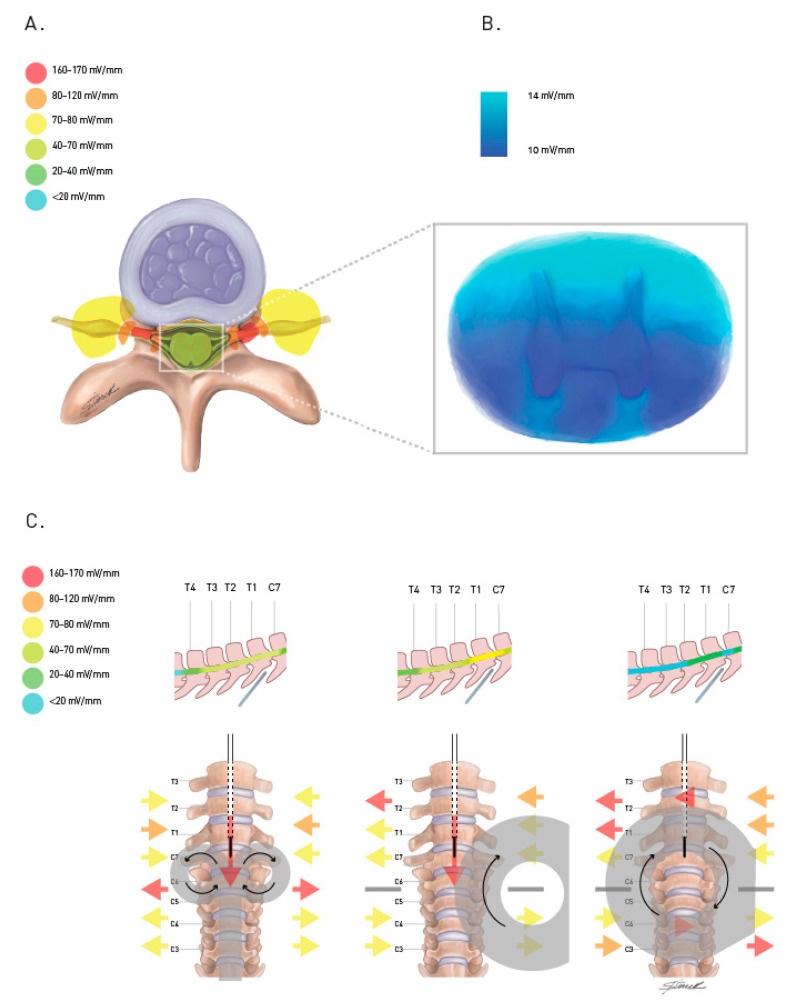
Transspinal magnetic stimulation induced electric fields, as suggested by the exploratory literature and computational models. (**A**). Generated current is concentrated on neuroforamina with estimated three to ten-fold greater intensity than at the spinal cord. (**B**). Current distribution in the spinal cord favors the posterior column, sparing the anterior horn and antero-lateral columns. (**C**). Generated current orientation on neuroforamina and spinal canal depends on coil type and positioning. Intraspinal transversal currents are substantially weaker than longitudinal currents when generated with similar coils. Root stimulation is optimal, leading to lower resting motor threshold and higher peak amplitudes when induced current is better aligned with the neuroforamina (orange and red arrows), preferably in an outwards direction (red arrow). mV/mm: millivolts per millimeter.

**Table 1 jcm-11-04748-t001:** Studies investigating clinical applications of TsMS.

Study	Population and Trial Design	Device, Location and Intensity	Frequency and Number of Sessions	Main findings
Nielsen et al., 1995 [34]	12 multiple sclerosis patients Open label trial	13.4 cm circular coil at T8 0.8 to 1.3 Tesla	12 Hz repetitive (5760 pulses) single session	Significant 28% reduction of stretch reflex amplitude and 50% increase of stretch reflex threshold. Significant reduction of Ashworth spasticity scale and daily activities questionnaire. Side effect of tight feeling, brief dizziness in one patient.
Nielsen et al., 1996 [35]	38 multiple sclerosis patients Double blinded randomized placebo-controlled trial	13.4 cm circular coil at T8 0.6 to 0.7 Tesla	25 Hz repetitive (10,000 pulses) or sham Two daily sessions for 7 days	Significant 27% reduction of stretch reflex amplitude in active group. Significant reduction of Ashworth spasticity scale and reflex grading only in the active group. Improvement of daily activities questionnaire in both groups. Significant baseline spasticity score between groups. Two episodes brief dizziness, one palpitation not confirmed by ECG after 2 h of stimulation
Nielsen et al., 1997 [32]	11 multiple sclerosis patients, 9 HC Open label trial	13.4 cm circular coil at T8 0.4, 0.6 or 0.8 Tesla	25 Hz (3750) repetitive, Single session Secondary comparison of 1 Hz, 10 Hz and 25 Hz 16 pulses.	Significant H reflex reduction at 1, 10, and 25 Hz in patients and controls after 16 pulses, but only in patients after full 3750 pulses session. No cortical motor evoked potentials changes Minor discomfort.
Krause et al., 2003 [36]	6 spinal cord injury patients (3 complete) Single blinded crossover-controlled trial	90 mm circular coil L3–L4 2 cm lateral 120% of lower limb contraction	20 Hz repetitive (2000 pulses) vs. electrical stimulation vs. sham Single session	Significant reduction in spastic tone in the Ashworth scale and modified Wartenberg’s pendulum test of spasticity after TsMS, similar to results produced by electrical stimulation, and superior to sham. No discomfort.
Krause et al., 2004 [37]	16 HC, 15 varied spinal lesion patients Open label trial	90 mm circular coil L3–L4 2 cm lateral 120% of lower limb contraction	20 Hz repetitive (2000 pulses) 1 session	Significant reduction in the Ashworth scale between 4 and 24 h after stimulation Significant increase in modified Wartenberg’s pendulum test of spasticity peak velocity in patients between 4 and 24 h but not HC. Effect was more pronounced contralateral to stimulation site. No reported side effects
Krause et al., 2005 [38]	Single spinal myelitis patient, single blinded controlled trial	90 mm circular coil L3–L4 2 cm lateral 120% of lower limb contraction	20 Hz, 15 Hz, 10 Hz (2000 pulses each), 3 sessions of each plus sham in random order	Significant reduction in relaxation index in all protocols compared to sham; no significant difference between frequency groups. No reported side effects
Niu et al., 2018 [39]	5 spinal cord injury patients (complete) requiring intermittent catheterization Crossover single blinded followed by open label	75 mm figure of eight coil at T12–L1 80% of paraspinal muscle contraction (0.8 to 1.2 Tesla)	Phase one: 1 Hz (720 pulses) vs. 30 Hz (21,600 pulses) one session Phase two: 1 Hz (720 pulses) in 16 weekly sessions	1 Hz was superior to 30 Hz in all patients in decreasing urethral pressure and increasing detrusor pressure in urodynamic test after phase one. Significant improvement in spontaneous urine volume, reduction in self catheterization frequency from 6.6 to 2.4 per day and increase in quality of life after phase two. No reported side effects
Tsai et al., 2009 [40]	22 spinal cord injury patients with constipation Open label trial	No coil description. coil At T9 and L3 1.1 to 1.54 Tesla	20 Hz, 800 pulses, at each site. Two daily sessions for 21 days	Significant improvement of colonic transit time (62.6 to 50.4 h) and bowel symptom questionnaires. Benefit was greater in questionnaires immediately after protocol completion but remained significantly greater than baseline after 3 months. No reported side effects
Chiu et al., 2009 [41]	16 patients with Parkinson’s disease and constipation Open label trial	No coil description. Coil at T9 and L3 1.1 to 1.54 Tesla	20 Hz, 800 pulses, at each site. Two daily sessions for 21 days	Significant improvement of colonic transit time (64.9 to 53.6 h), reduced residual barium and bowel symptom questionnaires. Sustained benefit in questionnaires after three months No reported side effects
Ari et al., 2014 [42]	37 Parkinson’s disease patients with camptocormia Randomized single blind crossover placebo-controlled trial	115 mm circular coil Level of worse angulation (thoracic or lumbar) 1 Tesla	5 Hz repetitive (40 pulses) or sham Single session	Reduction in camptocormia flexion angle by average 10.9°, no change with sham. Benefit duration was reported to be of three days, no difference observed after one week. Two patients reported mild lumbar discomfort lasting less than a day.
Menezes et al., 2020 [43]	5 Parkinson’s disease patients with freezing of gait. Open label trial	90 mm Circular coil at T5 90% of abdominal motor contraction	intermittent Theta burst (400 pulses) 3 sessions in one day	Significant improvement consisting of mean change of UPDRS of 6.6 points and freezing of gait questionnaire after 7 days of stimulation. No significant change of timed up and go was seen, or in any outcome immediately after stimulation or after 28 days. One patient reported transitory fatigue
Mitsui et al., 2022 [44]	100 Parkinson’s disease patients, Randomized single blinded placebo-controlled trial	Figure of eight coil at T12–L1 Intensity not explicit	5 Hz repetitive (50 pulses) or sham 8 biweekly sessions.	Significant improvement consisting of mean change of UPDRS of 10.28 points immediately after TsMS, 5.04 after three months and 2.38 at 6 months. Significant improvement in timed up and go test in all endpoints. Side effects were similar in active and sham groups and included self-limited back/arm pain, falls, and dizziness in a total of 11 patients.

HC: Healthy control UPDRS: Unified Parkinson’s Disease Rating Scale cm: centimeter. mm: millimeter. Hz: Hertz.

**Table 2 jcm-11-04748-t002:** Clinical data and neurophysiological responses of hereditary spastic paraplegia patients before and after the TsMS thoracic inhibitory protocol.

Clinical and Neurophysiological Data	Patient 1	Patient 2	Patient 3
Sex	Male	Female	Female
Age (years old)	40	37	63
Age of onset (years old)	1	26	50
Phenotype	Pure	Pure	Pure
Genotype	SPG4	SPG33	SPG33
Baseline SPRS score	21	25	34
Medications in use	none	none	none
Pre TsMS			
H_2_/H_1_ ratio with paired stimuli at ISI of 50 ms	1.71	1.13	1.34
H_2_/H_1_ ratio with paired stimuli at ISI of 100 ms	0.99	1.08	1.17
Post TsMS			
H_2_/H_1_ ratio with paired stimuli at ISI of 50 ms	0.10	0.09	0.05
H_2_/H_1_ ratio with paired stimuli at ISI of 100 ms	0.88	0.79	0.76

H_2_/H_1_: second H curve per first H curve ratio. ISI: interstimulus interval. ms: millisecond. SPG: spastic paraplegia. TsMS: transspinal magnetic stimulation. SPRS: Spastic Paraplegia Rating Scale.

**Table 3 jcm-11-04748-t003:** Clinical and neurophysiological evidence of spinal cord modulating effects of TsMS.

Clinical Response	Neurophysiological Response
Reduction in spasticity: Lower Ashworth scores Lower Reflex Grading scores	Electromyography testing: Reduces the amplitude of the H-reflex Reduces the amplitude and increases the threshold of the stretch reflex
Improvement in neurogenic bladder: Increase voluntary urination Reduce the number of catheterizations Improve questionnaires of bladder function	Urodynamic testing: Improve bladder capacity Improve urinary velocity
Improvement in neurogenic bowel: Reduces the time needed in bowel program Improve questionnaires of bowel symptoms	Colonic transit: Reduces colonic transit time
Improvement in Parkinson’s disease: Reduces freezing of gait Improves Camptocormia Improves parkinsonism scores	Still under study (unknown)

EMG: needle electromyography. TsMS: transspinal magnetic stimulation.

## Data Availability

All data relevant to this study were incorporated in this publication.

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
