# Peer review of "Controversies and Clinical Applications of Non-Invasive Transspinal Magnetic Stimulation: A Critical Review and Exploratory Trial in Hereditary Spastic Paraplegia"

_jcm, 2022, doi:10.3390/jcm11164748_

Round 1
Reviewer 1 Report
The Authors made an interesting hybrid article by reviewing up-to-date literature regarding the effects of transspinal magnetic stimulation as a possible treatment or therapeutic approach to various trauma and symptoms associated with neurodegenerative disorders. The manuscript is well written providing a novel insight into a somewhat neglected subject as transcranial magnetic stimulation is the focus of the research.
There are a few questions/suggestions:
- Authors might add a paragraph regarding the effects of repetitive transcranial magnetic stimulation in animal models with dominant spinal cord pathology EAE and provide a short rationale on how brain stimulation can lead to positive downstream effects
- Are there any suggestions on what spinal cord region should be stimulated? What are the criteria for the place of stimulation?
Author Response
Dear Reviewer
We thank you for the attention in reviewing our manuscript, highlighting sections not to standard and suggesting improvements.
Substantial change was made to the result section to better separate our interpretation and reviewed literature, as requested, leading to a slightly longer introduction and much longer discussion sections, but with no change whatsoever in our interpretation of reviewed research.
Additionally, a new paper on TsMS was published this month by Mitsui et al involving 100 patients with parkinson's disease, using a single blinded randomized paradigm. We added this important paper in our review section.
Below, we present our point by point answer to the raised questions and suggestions
REVIEWER 1
The Authors made an interesting hybrid article by reviewing up-to-date literature regarding the effects of transspinal magnetic stimulation as a possible treatment or therapeutic approach to various trauma and symptoms associated with neurodegenerative disorders. The manuscript is well written providing a novel insight into a somewhat neglected subject as transcranial magnetic stimulation is the focus of the research.
There are a few questions/suggestions:
- Authors might add a paragraph regarding the effects of repetitive transcranial magnetic stimulation in animal models with dominant spinal cord pathology EAE and provide a short rationale on how brain stimulation can lead to positive downstream effects
Answer: We thank the reviewer for the suggestion, and added a paragraph (line 400-409) as follows:
“Not only the mechanism of TsMS spinal cord modulation is not clear, but our review highlighted a significant question as spinal cord assessment via H reflex was mostly analyzed at the level of stimulation. Not only peripheral mechanisms could be more involved than spinal circuitry itself, but no information was obtained from different spinal levels. In a recent study with animal model of spinal injury 10 Hz TMS was able to not only to improve motor function, but to promote upregulation of pro-inhibitory membrane proteins and reduce the H reflex compatible with reduced motoneuron hyperexcitability [1], suggesting that spinal cord modulation is possible by descending modulation, at least from the cortex. We conducted an open label trial to address if this could also be possible from higher spinal levels using TsMS”
1. Wang S, Wang P, Yin R, Xiao M, Zhang Y, Reinhardt JD, Wang H, Xu G. Combination of repetitive transcranial magnetic stimulation and treadmill training reduces hyperreflexia by rebalancing motoneuron excitability in rats after spinal cord contusion. Neurosci Lett. 2022 Apr 1;775:136536. doi: 10.1016/j.neulet.2022.136536.
- Are there any suggestions on what spinal cord region should be stimulated? What are the criteria for the place of stimulation?
Answer: At the moment, clinical research has been extremely limited in repeated trials with different methodology, as most groups repeated the previous protocols in later studies, making it nearly impossible to adequately predict best settings for each pathology. Practically all previous clinical research concentrated on lower thoracic or lumbar regions, and most trials employed repetitive protocols and not theta burst, but for reasons individual to each group.
Therefore, at the moment, repetitive lower thoracic or lumbar high frequency repetitive stimulation appears to be the only protocol supported by any previous work for spasticity, constipation and parkinson's disease, and low frequency for urinary retention.
Our current work does hint at possible anti spastic properties of upper thoracic continuous theta burst protocols and our other pilot study used intermitent theta burst for parkinsons disease (menezes et al 2020), but both are very limited for clinical reasoning compared to the other reviewed literature in this manuscript.

Reviewer 2 Report
The paper needs some clarifications and revisions as listed below:
Major
· Methods, line 67: It is good stating that patients were walking at least 10 m without assistance. However, please also provide more information about the patients' other clinical features – specifically, the severity of their spasticity and the anti-spastic medications they use.
· Methods, line 69: Ref #8 is about electrical spinal cord stimulation for pain management. This reference does not fit the "Spastic Paraplegia Rating Scale (SPRS) and the H-reflex recovery curve" sentence.
· The Materials and Methods section noted that all evaluations were repeated after one month (line 70). Therefore, please provide the results obtained in one month.
· Please explain the rationale for selecting the T2 level for stimulation (line 74) in this study.
· Please elaborate on the method to determine the motor threshold of diaphragm contraction (line 76).
· In the methods section, line 89 states, "H-reflex recovery curves were obtained by paired stimuli at interstimulus intervals of 50, 100, 150, 200, 250, 300, 350, 400, and 500 ms." However, in the results section, only paired stimuli of 50 and 100 ms interstimulus intervals (ISI) are provided. Please provide an entire H-reflex recovery curve of all paired stimuli of performed ISI and discuss the results accordingly.
· Please clarify if the H-reflex recovery testing was performed once after the delivery of TsMS at 90 minutes or repeated immediately after the first and second TsMS at baseline?
· Provide references in Table 1.
· Please include patient groups, targeted pathology, TsMS location, and stim parameters, a summary of results in Table 1, and describe the details in the text. Thus, the readers to quickly see an adequate summary of reviewed articles.
Minor
· Please introduce PMS (Introduction section, line 42) acronym before using it in the text.
· Line 46: "Transcutaneous magnetic stimulation of the spine would also be an acceptable term but is longer." This sentence may be unnecessary.
· Table 3 does not consist of "μA." Please remove "μA: microampere" from the legend.
· Lines 354 and 372: square brackets are missing for ref # 48 and ref # 7
· A few typos (corrections):
Line 112: Teslas (Tesla)
Line 152: In in (In)
Table 2: lombar (lumbar)
Author Response
Dear Reviewer
We thank you for the attention in reviewing our manuscript, highlighting sections not to standard and suggesting improvements.
Substantial change was made to the result section to better separate our interpretation and reviewed literature, as requested, leading to a slightly longer introduction and much longer discussion sections, but with no change whatsoever in our interpretation of reviewed research.
Additionally, a new paper on TsMS was published this month by Mitsui et al involving 100 patients with parkinson's disease, using a single blinded randomized paradigm. We added this important paper in our review section.
Below, we present our point by point answer to the raised questions and suggestions
REVIEWER 2
This research is relatively new area, potentially providing with new insights on neural disorders including spinal cord. However, it is unclear for me whether this research objectly answers some research questions you mentioned in the introduction section (Lines 53-55). For example, no information was listed how to choose previous literatures. If these literatures were subjectively chosen, results would be subjective. Another thing is that the result section looks strange to me because this section includes your opinions and suggestions. I do not think that this is proper way when describing results. Furthermore, I do not think that this clinical study gives us some beneficial information due to insufficient information and without statistics. Finally, I put several comments in detail below.
Introduction
1) Lines 48-55: Although this introduction is written about TsMS, the background on your review is unclear. I think you need to explain it based on previous literatures.
Answer: We have revised our introduction and improved our introduction to offer a better background both to the review and our small exploratory trial.
2) Lines 48-55: What is the purpose combined with a literature review and clinical study with HSP patients? Some references have already reported the effects of TsMS on the spinal cord. Does this give us any new information?
Answer: We have made the reasoning for our exploratory trial clearer. It is clear from the literature that TsMS does affect the spinal cord and the best measure of effect so far has been through the H reflex as a measure of excitability. Previous literature however only offers insight in H reflex from the level of TsMS, usually from roots directly under the coil, therefore making it unclear if TsMS may affect distant levels or if it would have to be performed on the spinal cord level by level.
We used the exploratory trial to evaluate if stimulation at upper thoracic levels could change soleus H reflex and found very impressive change in the H reflex recovery suggestive of reduced hyperexcitability. While we plan a larger study involving also clinical outcomes, this result adds to the understanding of TsMS as all levels lower than stimulated might be affected.
The choice for patients with HSP was motivated by the fact that it is a frequent disease in movement disorders, neuromuscular and neurogenetic outpatient clinics, whose main symptom and clinical complaint is spasticity and besides being a model for spinal hyperexcitability, lacks effective treatment. While our trial was not focused on clinical benefit we plan on using it as a pilot before conducting larger follow up studies.
Materials and Methods
1) I think that article selection is one of the most important parts in extensive review, but no information how to select relevant articles, such as website and search term. To explain the therapeutic benefits, you objectively need to review according to previous literatures.
Answer: We agree with the reviewer and took care in altering our reported methodology to better detail how previous literature was selected, as described in line 73-82. Some redundant exploratory studies were omitted if presenting the same conclusion as previously reviewed literature, but no clinical trial was excluded. We would like to confirm that no negative clinical trials were found, but considering the limited number of articles a publishing bias can certainly not be ruled out.
2) Line 69: What is the reason measured H-reflex recovery curve?
Answer: In healthy subjects, a conditioning electrical stimulus leads to inhibition of the H-reflex. 1,2 The soleus H-reflex recovery curve, usually obtained by applying pairs of conditioning and test stimuli over the posterior tibial nerve in the popliteal fossa, shows early inhibition due to neurotransmitter depletion and motor neuron after hyperpolarization. 2 A brief period of inhibition is present in healthy subjects at short intervals, which was observed in our trial at ISI of 50 ms. It has been used in previous studies to evaluate spasticity. 3
- Gassel MM. A critical review of evidence concerning long-loop reflexes excited by muscle afferents in man. J Neurol Neurosurg Psychiatry 1970;33:358-362.
- Taborikova H, Sax DS. Conditioning of H-reflexes by a preceding subthreshold H-reflex stimulus. Brain 1969;92:203-212.
- Kumru H, Albu S, Valls-Sole J, Murillo N, Tormos JM and Vidal J. Influence of spinal cord lesion level and severity on H-reflex excitability and recovery curve. Muscle Nerve 2015;52:616-622.
3) Line 70: What does this protocol inhibit?
Answer: This was clarified, as continuous theta burst is used for cortical inhibitory protocols but there is no data supporting a likewise effect on spinal tissue. We now merely described the protocol as consisting of continuous theta burst stimulation - lines 91-92
4) Lines 74-75: In this study, TMS applied to the T2, but H-reflex was measured from the soleus muscle innervated by S1 and S2. Why was T2 chosen?
Answer: T2 was chosen as a level distant enough not to produce any spinal root response at lumbar or sacral roots. Higher levels could have been selected as the purpose of the trial was to study a distant effect from TsMS, but our group was previously familiar with thoracic stimulation and opted for a upper thoracic protocol. We clarified the purpose of T2 stimulation on the methodology line 96, and the purpose of our exploratory trial on the introduction lines 68-71 and discussion lines 400-409
5) Lines 75-76: Motor threshold was determined based on diaphragm contraction. Why was not soleus muscle chosen?
Answer: We selected T2 level in order to avoid sacral roots stimulation, therefore soleous muscle contraction was not expected. We did evaluate patients for any sign of muscle activity in lower limbs and found none, and presented this data in the results scetion
6) Lines 77-78: Is this cTBS protocol is a common way to induce neural plasticity on TsMS research? cTBS protocol usually consists of three-pulse bursts at 50 Hz repeated 200 ms for 40 s.
Answer: We thank you for pointing out the protocol description, and we clarified it. A total of 300 pulses were administered via standard cTBS protocol as the reviewer described, in three pulse bursts at 50hz totaling 20 seconds. We clarified the protocol description in the methodology sections, line 100
7) Line 88: Is 2 kHz-10 kHz correct?
Answer: The bandpass filter is 2 Hz to 10 kHz. We have rewritten it to clarify the sentence in line 110
8) Lines 88-89: What is the purpose used paired pulse for H-reflect measurement?
Answer: Paired pulses are necessary for obtaining the H reflex recovery curve, as they are used for conditioning and testing sequentially.
Results
1) I do not think that section 1 and 2 are results, and it looks discussion to me because these sections include your suggestions and opinions. I recommend you to separate these sections from the result.
Answer: We initially prepared a critical review with the goal of evaluating evidence as it was presented, however taking the reviewer suggestion and the risk of confounding the reader we opted to thoroughly follow the reviewer recommendation and separate the sections to allow for a fully objective result section. This led to a large revision of the text but conclusions and presented evidence were unchanged.
We believe the end result is much superior and thank the reviewer for the recommendation.
2) Lines 100-103: Are these noninvasive stimulation methods for the spinal cord or brain?
Answer:We made the paragraph clearer in explaining that those are cortical effects, and moved the paragraph to the introduction. We believe these modulatory paradigms are needed for the comprehension of protocol selection rationale in spinal cord studies, even without any substantial evidence of equivalent effect in the latter.
3) Lines 110-117: I am not sure whether you can insist that TsMS is very safe. Although up to 2.4 Teslas did not induce cardiac arrhythmias, does this evidence generalize for humans?
Answer: The referenced study with animals was used to study the safety of direct magnetic fields administered close to the heart. We agree that while this is not sufficient evidence of TsMS safety in humans, this data addresses the main early concern of the technique over thoracic regions. We clarified that this study only answered some early concerns.
We do believe TsMS is probably safe with conventional devices, as no serious side effects were ever reported in either exploratory or clinical studies in humans. Data is limited however on patients with pacemakers or epilepsy, as studies explicitly excluded these patients.
4) Lines 121-122: This sentence does not make sense for me.
Answer: We clarified the sentence, as follows: “Most exploratory studies in humans were conducted in the early years of magnetic stimulation. Several authors identified that produced CMAP has a fixed latency regardless of the coil position or intensity, even when moving the coil away from the stimulation site in any direction laterally or rostro caudally up to the point where it is no longer elicitable [14,15,16], and conclude that stimulation would occur at a fixed point for each studied muscle, therefore unlikely to be from direct corticospinal tract activation and possibly at an area of high current concentration such as the neuroforamina.”
5) Lines 140-145: What is the purpose of this paragraph? This is not related to spinal cord stimulation.
Answer: Although the discussed literature had the purpose of foramen magnum stimulation, the authors employed not only coils centered at the inion but also at high cervical levels, with similar successful foramen magnum magnetic stimulation. We believe expliciting this exploration was necessary considering the possible contribution of this review in guiding future research, helping authors planning stimulation at cervical levels to consider other effects other than TsMS. This data could also be helpful to interpret findings of descending tract stimulation in a single study employing high cervical TsMS[1]. We however rewrote the paragraph to better fit the rest of the section (lines 142-159 ).
- Tomberg C. Transcutaneous magnetic stimulation of descending tracts in the cervical spinal cord in humans. Neuroscience letters. 1995;188(3), 199-201.
6) Lines 166-170: This paragraph does not make sence for me what you want to insist.
Answer: This paragraph describes the most recent exploratory TsMS study in humans. Although most of the research contribution is in detailing CMAP elicited by TsMS electrophysiological components, arguably out of the scope of this review, it demonstrates very similar findings to previous literature despite decades of progress in magnetic stimulation. As the objective of our review was to provide a comprehensive analysis of literature we believe it is a necessary study to include.
7) Line 186: How the model was made? Also, is this your data?
Answer: The figure summarizes data from previous exploratory studies, integrating data obtained from exploration with human spinal columns and computational model. While it does not include new data, the figure is an original creation for this paper. As it does not fit the result section, it was moved to the discussion section at page 10.
8) Lines 226-229: I could not understand the general effects of repetitive TsMS on spinal cord excitability. You mentioned above that high repetitive TMS induces facilitation and vice versa, but this sentence shows opposite results.
Answer: We highlighted the lack of validity of cortical TMS protocols in spinal circuitry in lines 329-334. While clinically low frequency appeared to be better for bladder function in one study[1] and high frequency for spasticity in another[2], more studies are required to adequately understand what protocols are really inhibitory or excitatory in spinal cord circuitry.
- Knikou M. Neurophysiological characteristics of human leg muscle action potentials evoked by transcutaneous magnetic stimulation of the spine. Bioelectromagnetics. 2013 Apr;34(3):200-10. doi: 10.1002/bem.21768.
- Niu T, Bennett CJ, Keller TL, Leiter JC, Lu DC. A Proof-of-Concept Study of Transcutaneous Magnetic Spinal Cord Stimulation for Neurogenic Bladder. Sci Rep. 2018 Aug 22;8(1):12549. doi: 10.1038/s41598-018-30232-z.
9) Lines 319-324: This paragraph is arbitrary. In spite of you measured some ISIs, only two ISIs were shown as results.
Answer: We chose to expose only the 50 and 100 ms ISI for didactic purposes, given that our objective was to assess whether the TsMS protocol had any effect and from 100 ms onwards the H reflex has already returned to close to baseline in the evaluated patients, as shown in figure 2. Thus, there would be no significant change in the interpretation of our findings and discussion.
We however agreed that the paragraph was arbitrary and removed it, including the observation that the rest of the measured isis returned to baseline at 100ms. All measured H2/H1 ratios after 200ms in both pre and post TsMS were 1.0, as usual in H reflex recovery curve.
10) Line 334: ''There was no significant difference'': no statistical information.
Answer: Thank you. We corrected the term, changing it to “no evident difference”.
11) Where is one month later results?
Answer: While initially we planned on expanding the current work with long term findings, we subsequently considered they would best fit a subsequent larger study including clinical outcomes and longitudinal followup and a larger sample allowing for statistical analysis. We have thus excluded the one month mention from the manuscript as the purpose of this article was to show the acute effects of TsMS on H reflex. if the reviewer believe it to be of interest we will be happy to include these results.
Discussion
1) Lines 346-349: You are about explaining the effect of TsMS on the spinal cord excitability. However, next paragraph explains about peripheral nerve magnetic stimulation on H-reflect and F-waves. I cannot understand the meaning of this paragraph.
Answer: We took the opportunity of removing subjectivity from the results section to rewrite confusing segments of the discussion. We believe it’s now clearer.
2) Line 376: You should explain what H-reflex recovery means.
Answer: We agree and included an explanation on H reflex recovery curve at lines 57-61
3) Lines 381-382: I do not understand what different ISIs mean.
Answer: As the ISI is extended, the neuronal hyperpolarization is reduced and the ease of depolarization increased, leading to a progressive waxing in the amplitude of the H- reflex to the baseline value.
4) Line 417: Although you mentioned no data are available on patients with some neural disorders, can you say TsMS is very safe?
Answer: As no serious side effects were reported from the pool of investigated studies, involving 222 neurological patients and 199 healthy controls, we believe the technique to be indeed safe, however we agreed to remove the adjective “very”, as high intensity was seldom tested and some pathologies were excluded from studies. We added these considerations in lines 431-444
We took the opportunity of the revision to improve the clinical studies section and included a table describing all identified literature (table 1), including all side effects and tolerability description in the results column, which should complement the supplementary table 1 reviewing exploratory trials further supporting our claim that the technique is safe.

Reviewer 3 Report
Sammary)
This article investigated the effect of TsMS on the spinal cord excitability through a narrative review and exploratory pilot study with three HSP patients. Although it still needs more evidence on TsMS, any severe side effects have not been reported so far. TsMS might become a potential neurorihabilitation tool improving spasticity, freezing gate, and hyperreflexia.
General comments)
This research is relatively new area, potentially providing with new insights on neural disorders including spinal cord. However, it is unclear for me whether this research objectly answers some research questions you mentioned in the introduction section (Lines 53-55). For example, no information was listed how to choose previous literatures. If these literatures were subjectively chosen, results would be subjective. Another thing is that the result section looks strange to me because this section includes your opinions and suggestions. I do not think that this is proper way when describing results. Furthermore, I do not think that this clinical study gives us some beneficial information due to insufficient information and without statistics. Finally, I put several comments in detail below.
Introduction)
1) Lines 48-55: Although this introduction is written about TsMS, the background on your review is unclear. I think you need to explain it based on previous literatures.
2) Lines 48-55: What is the purpose combined with a literature review and clinical study with HSP patients? Some references have already reported the effects of TsMS on the spinal cord. Does this give us any new information?
Materials and Methods
1) I think that article selection is one of the most important parts in extensive review, but no information how to select relevant articles, such as website and search term. To explain the therapeutic benefits, you objectively need to review according to previous literatures.
2) Line 69: What is the reason measured H-reflex recovery curve?
3) Line 70: What does this protocol inhibit?
4) Lines 74-75: In this study, TMS applied to the T2, but H-reflex was measured from the soleus muscle innervated by S1 and S2. Why was T2 chosen?
5) Lines 75-76: Motor threshold was determined based on diaphragm contraction. Why was not soleus muscle chosen?
6) Lines 77-78: Is this cTBS protocol is a common way to induce neural plasticity on TsMS research? cTBS protocol usually consists of three-pulse bursts at 50 Hz repeated 200 ms for 40 s.
7) Line 88: Is 2 kHz-10 kHz correct?
8) Lines 88-89: What is the purpose used paired pulse for H-reflect measurement?
Results
1) I do not think that section 1 and 2 are results, and it looks discussion to me because these sections include your suggestions and opinions. I recommend you to separate these sections from the result.
2) Lines 100-103: Are these noninvasive stimulation methods for the spinal cord or brain?
3) Lines 110-117: I am not sure whether you can insist that TsMS is very safe. Although up to 2.4 Teslas did not induce cardiac arrhythmias, does this evidence generalize for humans?
4) Lines 121-122: This sentence does not make sense for me.
5) Lines 140-145: What is the purpose of this paragraph? This is not related to spinal cord stimulation.
6) Lines 166-170: This paragraph does not make sence for me what you want to insist.
7) Line 186: How the model was made? Also, is this your data?
8) Lines 226-229: I could not understand the general effects of repetitive TsMS on spinal cord excitability. You mentioned above that high repetitive TMS induces facilitation and vice versa, but this sentence shows opposite results.
9) Lines 319-324: This paragraph is arbitrary. In spite of you measured some ISIs, only two ISIs were shown as results.
10) Line 334: ''There was no significant difference'': no statistical information.
11) Where is one month later results?
Discussion
1) Lines 346-349: You are about explaining the effect of TsMS on the spinal cord excitability. However, next paragraph explains about peripheral nerve magnetic stimulation on H-reflect and F-waves. I cannot understand the meaning of this paragraph.
2) Line 376: You should explain what H-reflex recovery means.
3) Lines 381-382: I do not understand what different ISIs mean.
4) Line 417: Although you mentioned no data are available on patients with some neural disorders, can you say TsMS is very safe?
Author Response
Dear Reviewer
We thank you for the attention in reviewing our manuscript, highlighting sections not to standard and suggesting improvements.
Substantial change was made to the result section to better separate our interpretation and reviewed literature, as requested, leading to a slightly longer introduction and much longer discussion sections, but with no change whatsoever in our interpretation of reviewed research.
Additionally, a new paper on TsMS was published this month by Mitsui et al involving 100 patients with parkinson's disease, using a single blinded randomized paradigm. We added this important paper in our review section.
Below, we present our point by point answer to the raised questions and suggestions
REVIEWER 3
The paper needs some clarifications and revisions as listed below:
Major
- Methods, line 67: It is good stating that patients were walking at least 10 m without assistance. However, please also provide more information about the patients' other clinical features – specifically, the severity of their spasticity and the anti-spastic medications they use.
Answer: We included patient spasticity score and medications as requested in text (lines 300-301) and table 2. Patients had Spastic Paraplegia Rating Scale of 21, 25 and 34 and used no medication whatsoever, but were adherent to physical therapy.
- Methods, line 69: Ref #8 is about electrical spinal cord stimulation for pain management. This reference does not fit the "Spastic Paraplegia Rating Scale (SPRS) and the H-reflex recovery curve" sentence.
Answer: We corrected the reference and thank the reviewer for pointing out the mistake. The correct reference supporting how H reflex recovery curve was performed is as follows:
Kumru H, Albu S, Valls-Sole J, Murillo N, Tormos JM, Vidal J. Influence of spinal cord lesion level and severity on H-reflex excitability and recovery curve. Muscle Nerve . 2015 Oct;52(4):616-22. doi: 10.1002/mus.24579.
- The Materials and Methods section noted that all evaluations were repeated after one month (line 70). Therefore, please provide the results obtained in one month.
Answer: While initially we planned on expanding the current work with long term findings, we subsequently considered they would best fit a subsequent larger study including clinical outcomes and longitudinal followup and a larger sample allowing for statistical analysis. We have thus excluded the one month mention from the manuscript as the purpose of this article was to show the acute effects of TsMS on H reflex. if the reviewer believe it to be of interest we will be happy to include these results.
- Please explain the rationale for selecting the T2 level for stimulation (line 74) in this study.
Answer: T2 was chosen as a level distant enough not to produce any spinal root response at lumbar or sacral roots. Higher levels could have been selected as the purpose of the trial was to study a distant effect from TsMS, but our group was previously familiar with thoracic stimulation and opted for an upper thoracic protocol. We clarified the purpose of T2 stimulation on the methodology line 97, and the purpose of our exploratory trial on the introduction lines 70-71 and discussion lines 403-412
- Please elaborate on the method to determine the motor threshold of diaphragm contraction (line 76).
Answer: We clarified the methodology. Diaphragm contraction was monitored by hand by a second examiner while stimulation intensity was gradually increased.
- In the methods section, line 89 states, "H-reflex recovery curves were obtained by paired stimuli at interstimulus intervals of 50, 100, 150, 200, 250, 300, 350, 400, and 500 ms." However, in the results section, only paired stimuli of 50 and 100 ms interstimulus intervals (ISI) are provided. Please provide an entire H-reflex recovery curve of all paired stimuli of performed ISI and discuss the results accordingly.
Answer: We chose to expose only the 50 and 100 ms ISI for didactic purposes, given that our objective was to assess whether the TsMS protocol had any effect and from 100 ms onwards the H reflex has already returned to close to baseline in the evaluated patients, as shown in figure 2. Thus, there would be no significant change in the interpretation of our findings and discussion.
We however agreed that the paragraph was arbitrary and removed it, including the observation that the rest of the measured isis returned to baseline at 100ms. All measured H2/H1 ratios after 200ms in both pre and post TsMS were 1.0, as usual in H reflex recovery curve.
- Please clarify if the H-reflex recovery testing was performed once after the delivery of TsMS at 90 minutes or repeated immediately after the first and second TsMS at baseline?
Answer: Neurophysiological evaluations were performed at the baseline and once after the second delivery of TsMS. We have rewritten it to clarify the sentence in lines 91-92
- Provide references in Table 1.
- Please include patient groups, targeted pathology, TsMS location, and stim parameters, a summary of results in Table 1, and describe the details in the text. Thus, the readers to quickly see an adequate summary of reviewed articles.
Answer: Table 1 now includes all parameters, population and study details as requested, and we believe it greatly improved the clinical results section and are grateful for the suggestion.
Minor
- Please introduce PMS (Introduction section, line 42) acronym before using it in the text.
Answer: We thank the reviewer for the correction.
- Line 46: "Transcutaneous magnetic stimulation of the spine would also be an acceptable term but is longer." This sentence may be unnecessary.
Answer: We removed the sentence as suggested.
- Table 3 does not consist of "μA." Please remove "μA: microampere" from the legend.
Answer: We removed the unnecessary term from the legend.
- Lines 354 and 372: square brackets are missing for ref # 48 and ref # 7
Answer: We thank the reviewer for the correction.
- A few typos (corrections):
Line 112: Teslas (Tesla)
Line 152: In in (In)
Table 2: lombar (lumbar)
Answer: We made the corrections.
Round 2
Reviewer 3 Report
They properly answered my concerns, and I do not have any comments.